# Ca^2+^ Signaling and Its Potential Targeting in Pancreatic Ductal Carcinoma

**DOI:** 10.3390/cancers13123085

**Published:** 2021-06-21

**Authors:** Louay Bettaieb, Maxime Brulé, Axel Chomy, Mel Diedro, Malory Fruit, Eloise Happernegg, Leila Heni, Anaïs Horochowska, Mahya Housseini, Kekely Klouyovo, Agathe Laratte, Alice Leroy, Paul Lewandowski, Joséphine Louvieaux, Amélie Moitié, Rémi Tellier, Sofia Titah, Dimitri Vanauberg, Flavie Woesteland, Natalia Prevarskaya, V’yacheslav Lehen’kyi

**Affiliations:** 1Option Biology, Master Biology-Health, Faculty of Medicine Henry Warembourg, University of Lille, 59120 Loos, France; louay.bettaieb.etu@univ-lille.fr (L.B.); maxime.brule.etu@univ-lille.fr (M.B.); axel.chomy.etu@univ-lille.fr (A.C.); mel.diedro.etu@univ-lille.fr (M.D.); malory.fruit.etu@univ-lille.fr (M.F.); eloise.happernegg.etu@univ-lille.fr (E.H.); leila.heni.etu@univ-lille.fr (L.H.); anais.horochowska.etu@univ-lille.fr (A.H.); mahya.housseini.etu@univ-lille.fr (M.H.); kekely.klouyovo.etu@univ-lille.fr (K.K.); agathe.laratte.etu@univ-lille.fr (A.L.); alice.leroy3.etu@univ-lille.fr (A.L.); paul.lewandowski.etu@univ-lille.fr (P.L.); josephine.louvieaux.etu@univ-lille.fr (J.L.); amelie.moitie.etu@univ-lille.fr (A.M.); remi.tellier.etu@univ-lille.fr (R.T.); sofia.titah.etu@univ-lille.fr (S.T.); dimitri.vanauberg.etu@univ-lille.fr (D.V.); flavie.woesteland.etu@univ-lille.fr (F.W.); 2Laboratory of Cell Physiology, INSERM U1003, Laboratory of Excellence Ion Channels Science and Therapeutics, Department of Biology, Faculty of Science and Technologies, University of Lille, 59650 Villeneuve d’Ascq, France; natalia.prevarskaya@univ-lille.fr; 3University Lille, Inserm, U1003-PHYCEL-Physiologie Cellulaire, F-59000 Lille, France

**Keywords:** pancreatic cancer, PDAC, Ca^2+^, Ca^2+^ channels, Ca^2+^ signaling, PDAC treatment

## Abstract

**Simple Summary:**

Pancreatic adenocarcinoma is the fourth most common cause of cancer-related death in the world and is estimated to be the second cause of cancer deaths by 2030. Its complicity of treatment is explained by unique physiology and microenvironment of the exocrine pancreas, where calcium plays an essential role. This review is focused on the Ca^2+^-dependent proteins and signalization pathways studied so far in pancreatic adenocarcinoma. Ca^2+^ signaling is suggested as potential and prospective target in pancreatic carcinoma treatment and drug resistance.

**Abstract:**

Pancreatic cancer (PC) is a major cause of cancer-associated mortality in Western countries (and estimated to be the second cause of cancer deaths by 2030). The main form of PC is pancreatic adenocarcinoma, which is the fourth most common cause of cancer-related death, and this situation has remained virtually unchanged for several decades. Pancreatic ductal adenocarcinoma (PDAC) is inherently linked to the unique physiology and microenvironment of the exocrine pancreas, such as pH, mechanical stress, and hypoxia. Of them, calcium (Ca^2+^) signals, being pivotal molecular devices in sensing and integrating signals from the microenvironment, are emerging to be particularly relevant in cancer. Mutations or aberrant expression of key proteins that control Ca^2+^ levels can cause deregulation of Ca^2+^-dependent effectors that control signaling pathways determining the cells’ behavior in a way that promotes pathophysiological cancer hallmarks, such as enhanced proliferation, survival and invasion. So far, it is essentially unknown how the cancer-associated Ca^2+^ signaling is regulated within the characteristic landscape of PDAC. This work provides a complete overview of the Ca^2+^ signaling and its main players in PDAC. Special consideration is given to the Ca^2+^ signaling as a potential target in PDAC treatment and its role in drug resistance.

## 1. Introduction

The exocrine pancreas is an organ containing acinar cells essential for the production of digestive enzymes required for the assimilation of nutriments [1]. In pancreatic cancer (PC), the growing tumor exerts pressure on the common bile duct, preventing the bile from being released and thus leading to jaundice, permanent fatigue, weight loss, clear stools, diarrhea, and pain, especially at the epigastric level. These symptoms occur in the late stage of the disease, and in 90% of cases to PDAC [2].

Currently, the risk of developing PC is 1.49%, i.e., one in 67 people, and this disease is recognized only at an advanced stage in people from 40 to 85 years old with a maximum incidence from 60 years old [2]. The incidence varies with the ethnic origin, being higher in industrialized regions, such as Europe, North America and Oceania; and in contrast, being lower in developing continents as Africa [3].

According to the current prognosis, being ranked in fourth place in cancer-related deaths globally, this disease tends to spread and is expected to reach second place by 2030 [3]. In fact, 432,232 new deaths were recorded in 2018 emphasized by the survival rate, which went from 6% to 9% from 2014 to 2018 [4]. After the treatment, the one-year survival rate is 24% and decreases by 9% for five years [3].

Moreover, PDAC represents 90% of the pancreatic cancers. The other 10% of pancreatic cancer, such as pancreatic acinar cell carcinoma or solid pseudopappilary neoplasm, are less frequent, and thus, less studied. Few scientists studied those cancers, such as Ohuchida et al. who showed a difference in the S100A2 expression level, according to the pancreatic cancer type, as intraductal papillary mucinous neoplasm or pancreatic intraepithelial neoplasia [5]. Yet, a small amount of knowledge on the calcium-linked mechanisms in the other pancreatic cancers exists so far, and needs further investigations.

Pancreatic ductal adenocarcinoma (PDAC) is inherently linked to the unique physiology and microenvironment of the exocrine pancreas, such as pH, mechanical stress and hypoxia. Whereas, Ca^2+^ signals are essential to detect and integrate signals from the microenvironment, thereby becoming relevant in cancer. Ca^2+^ levels are altered by mutations or an aberrant expression of key proteins, leading to the deregulation of Ca^2+^-dependent effectors that control signaling pathways. Therefore, cell behavior will be altered in a way to promote proliferation, survival and invasion, which are all the hallmarks of cancer [6].

Today, doctors are able to issue a diagnosis, staging, tumor size, node, metastasis (TNM) on a PDAC, and in particular, using computed tomography [7]. However, most of them are detected at late stages of cancer and the chances of recovery are minimal [8]. Therefore, the early detection and a better knowledge of the mechanisms will make it possible to understand Ca^2+^ signaling in PDAC and its main players. This could yield a potential class of targets in PDAC treatment with their role in drug resistance.

## 2. Ca^2+^ Transporters in PDAC

Ca^2+^ transporters are important in many pathological processes, notably in cancer where they can regulate some cancer hallmarks. These transporters have a key role in calcium homeostasis through their ability to generate bidirectional Ca^2+^ flows (in and out of the cell). There are three classes of calcium transporters: Channels, pumps and other transporters like exchangers. The specific feature of Ca^2+^ transporters is partly defined by their structure and by extrinsic factors [9]. Each category of transporters will be detailed in the following sections and is listed in the Table 1.

### 2.1. Channels

Ion channels are proteins, which are located on the plasma membrane (PM) and are found in all types of tissues. They ensure rapid transport of ions between the extracellular and the intracellular compartments. These ion channels are able to open and close; and while being open there is a distribution of ions, which will modify the electrical and chemical properties of the cell leading to different cellular processes. Regulating the opening and closing of ion channels is important for maintaining Ca^2+^ homeostasis and the induction of cellular processes, such as proliferation, apoptosis and migration. Deregulation in the expression of these ion channels could lead to the manifestation of various pathologies, such as cancer [20].

There are some Ca^2+^ channels discovered in PDAC, but their roles are still poorly understood. Among Ca^2+^ channel families shown to be involved in PDAC, there is a transient receptor potential (TRP) channels super-family, which is a family of 28 mammalian ion channel proteins [21], the majority of which are Ca^2+^ permeable, apart from TRP cation channel subfamily M member 5 and 6 (TRPM5 and TRPM6). This family is divided into sub-families, which are as follows: TRP canonical (TRPC), being involved in the exocytosis or in the stimulation of the secretion, like TRPC1 which is localized at the lateral membrane and is able to regulate the proliferation of PDAC cells via TGF-β signaling. As concerns TRPC3 and TRPC6 channels whose localization is still unknown, they may be involved like the others, either in the exocytosis or in the stimulation of the secretion. There are TRPM channels, as the TRPM7 channel, which is a Ca^2+^/Mg^2+^ channel fused with a functional kinase domain belonging to the α-kinase family. The TRPM7 is involved in the migration and/or the invasion of epidermal cancer cells, including neuroblastoma, glioblastoma, PDAC, etc., and is localized at the luminal level of the membrane [22]. There is also a vanilloid subfamily (TRPV), like TRPV6, whereby location and function are not known. Finally, there are TRP ankyrin-like (TRPA), TRP polycystin (TRPP) and TRP mucolipin (TRPML) families of channels. These TRPs are activated by different stimuli, like heat, and regulate many physiological processes [23].

On the other hand, there is also a store-operated channel (SOC) family. The members and regulators of the SOC family of the proteins are known as Ca^2+^ release-activated Ca^2+^ channel modulator 1, 2 and 3 (ORAI1, ORAI2 and ORAI3) and stromal interaction molecule 1 (STIM1). ORAI1 and STIM1 are located at the basolateral pole on the PM, and on the endoplasmic reticulum (ER), respectively. They have probably a role in the exocytosis or in the secretion while contributing to carcinogenesis and are involved in the regulation of proliferation of PDAC cells [24]. There are also ORAI2 and ORAI3 which have an unknown role while ORAI3 has rather basolateral localization.

In addition, there are the families of voltage-gated Ca^2+^ channels, bipolar channels, ryanodine receptors and finally the inositol triphosphate (IP3) family in the ER, which are important in maintaining Ca^2+^ homeostasis [25].

In cancer cells, the entry of Ca^2+^ mainly occurs through the SOC family, but also through TRPs, which are selective for Ca^2+^ and Na^+^ [10]. Ca^2+^ signaling is linked to the secretion of exocrine pancreatic fluid because acinar and ductal cells in the pancreas are regulated by receptors that modify the concentration of Ca^2+^. The Ca^2+^ signal is initiated by acetylcholine (ACh), while binding to specific receptors, which generate specific Ca^2+^ signals. This pathway begins with Ca^2+^-activating phospholipase C (PLC), which hydrolyzes phospholipids, yielding IP3 and diacylglycerol (DAG). The IP3 binds to its receptors located in the ER of acinar cells. As such, Ca^2+^ stores are depleted, which results in a clustering of STIM1, which activates the SOCs following by TRP channels, leading to an influx of Ca^2+^.

### 2.2. Pumps

In numerous cancers, including PDAC, the Warburg effect represents a tumor cell metabolism shift towards glycolysis, despite the abundance of O_2_. This aberrant metabolic profile confers tumor survival advantages and allows the production of intermediates for macromolecule biosynthesis essential for proliferation [15]. Furthermore, the intervention of characteristic enzyme isoforms in this metabolism, like pyruvate kinase isozyme M2 (PKM2) helps to maintain the important production of ATP vital for fueling key energy-dependent processes [26].

One of those modifications is the Ca^2+^ efflux via the ATP-dependent PM Ca^2+^-ATPase (PMCA), which is the predominant Ca^2+^ efflux pathway in human PDAC. It takes part in the regulation of Ca^2+^ homeostasis by maintaining a low intracellular Ca^2+^ concentration (around 100 nM), and thus, preventing the cell from death [15]. There are four isoforms of PMCAs (PMCA1–4), encoded by four different genes (ATP2B1–4). However, the alternate splicing of this gene mRNA transcripts results in a diverse array of more than 30 splice variants. These isoforms are expressed in different areas of the body: PMCA2–3 are “rapid response pumps” expressed in excitatory cells like muscle cells; PMCA1–4 are ubiquitous throughout all tissues in humans, while PMCA4 seems to have a specific role in proliferative signaling, NF-κB nuclear translocation, tumor necrosis factor (TNF)-induced cell death and migration, which all are important pathways for the cancer hallmarks conservation [17].

A study [15] performed on PDAC cells (PANC-1 and MIA PaCa-2 ATCC) showed that the inhibition of glycolysis in these cell lines leads to ATP depletion, PMCA inhibition and an irreversible intracellular Ca^2+^ concentration overload. By reversing the PDAC cells metabolism, while depriving cells of glucose and supplementing with α-ketoisocaproate or galactose (substrates that promote mitochondrial metabolism) high glycolytic PDAC cells dependence is established, particularly the glycolytic dependence of the PMCA. This feature could be exploited therapeutically to treat PDAC.

Further, the role of PMCAs, especially PMCA1, has been discovered in the interaction with the transporter CD147 [16], upregulated in some cancers, including PDAC, which acts as an ancillary protein by protecting it against degradation and contributing to its membrane translocation. Indeed, the inhibition of this transporter results in modifications in Ca^2+^ homeostasis and in some vital cellular processes such as cell cycle arrest, and thus, cell growth.

In addition, the same authors demonstrated that KM2, which provides ATP to the PMCA, is oncogenic and overexpressed in PDAC [26], thereby increasing the PMCA activity and reducing the sensibility to cell death induced by the PMK2 inhibitor shikonin (SHK). The use of this inhibitor has also revealed that PMK2 boosts cell proliferation, cell migration and limits the cell death caused by the inhibition of glycolysis, ATP depletion, inhibition of PMCA and cytotoxic Ca^2+^ overload. Finally, the authors concluded that depleting this pool of ATP for the PMCA provided by PKM2 can represent a novel therapeutic strategy to treat PDAC.

A recent study [17] has explored more accurately the role of the PMCA isoform PMCA4, which is definitively overexpressed in PDAC and is linked to poor patient survival. Researchers used MIA PaCa-2 cells, which almost exclusively express PMCA4, and knocked-downed (KD) the PMCA4 expression using siRNA. This KD resulted in a reduction of cytosolic Ca^2+^ clearance, cell migration and in a PDAC cell sensitivity to apoptosis. All these data suggest that the expression of PMCA4 promotes cell migration and apoptotic resistance in PDAC cells. In such a way, PMCA4 may constitute a novel therapeutic target in PDAC by sensitizing PDAC cells to apoptosis in a specific way.

### 2.3. Transporters

Transporters maintain the balance of Ca^2+^ levels between the rest state and cell activity by ensuring the movement of Ca^2+^ through the membranes in and out of the cell, and its intracellular storage at the level of organelles under normal conditions. There are two types of transport: Passive/facilitated diffusion and active. The first does not require energy because it is done in the direction of the gradient. While, the second uses ATP to move Ca^2+^ and unlike the first one, it is done against the gradient of concentration. Depending on their mechanics there are 3 classes: Ca^2+^ channels, pumps and exchangers [9]. It has been showed that Ca^2+^ transporters are involved in several cancers, e.g., Na^+^/Ca^2+^ exchanger (NCX), which is a membrane transporter that can exchange one Ca^2+^ out of the cell for every 3 Na^+^ into the cytoplasm or in the opposite direction under special conditions [27]. There are 3 isoforms identified in humans (NCX1, NCX2, NCX3) associated with the tumor progression, but their role in PDAC still have to be understood [28].

Moreover, the ER also has an important role in many cell metabolism processes and it is a Ca^2+^ storage organelle. Ca^2+^ homeostasis is vital for cell survival and any imbalance will lead to ER stress that would activate unfolded-protein response (UPR) and pro-apoptotic signaling controlled by Sarco/ER Ca^2+^-ATPases (SERCA) [18]; a pump that transports Ca^2+^ from the cytosol into the sarcoplasmic reticulum against the concentration gradient. In a recent study concerning SERCA2, a carbazole alkaloid (mahanine) was used to induce ER-stress to analyze its impact and Ca^2+^ signaling cross-talk on the survival of PDAC cells, the results revealed that mahanine increased intracellular Ca^2+^ leakage from ER via time-dependent manner and initiated UPR pathway with the upregulation of SERCA2 isoform [29]. It highlights the importance of time-dependent changes in intracellular Ca^2+^ homeostasis, and emphasizes the altered activity of Ca^2+^-dependent molecular pathway in the treated cells. Similarly, SERCA inhibitors can also induce apoptosis by decreasing [Ca^2+^]_ER_ triggering UPR in cancer [30].

Mitochondria participates in cellular Ca^2+^ signaling using its transporters: Mitochondrial calcium uniporter (MCU) and mitochondrial NCX, respectively. MCU is sensitive to Ca^2+^ changes concentration in mitochondrial intermembrane space and can contribute to the development of some cancers such as PDAC [31]. Based on the awareness that MCU ensures the transport of Ca^2+^ from the cytoplasm to the mitochondrial matrix, a single-pass membrane protein with the aspartate rich tail 1 (SMDT-1) protein, which is a regulatory subunit of MCU, is a Ca^2+^ sensor that can protect mitochondria from Ca^2+^ depletion by coupling Ca^2+^ sensors and MCU on both sides of the inner mitochondrial membrane [14].

Secretory pathway Ca^2+^-ATPase (SPCA) localized in Golgi apparatus, can transport Ca^2+^ and Mn^2+^ from the cytoplasm into Golgi and post-Golgi vesicles. Two isoforms exist: SPCA1 and SPCA2. The second isoform is associated with the absorptive and secretory epithelia as observed in pancreas, salivary and mammary glands. There is a study demonstrated that SPCA2 interacts with ORAI1 to restore Ca^2+^, but like NXC, it is still uninvestigated in PDAC [32].

All these transporters of calcium are most likely to play a crucial role in PDAC, and need to be considered in conjunction with the downstream pathways PDAC cells employ to gain the phenotypic advantages required for tumorigenesis. The graphic presentation of these Ca^2+^ transporters at the sub-cellular level is indicated in the Figure 1.

Ca^2+^ entry after the activation of these transporters will induce a signal transduction, which has a major role in cell processes as cell mobility, adhesion or migration by regulating signaling pathways of proliferation, apoptosis and autophagy. Many molecules, including kinases are involved in triggering signaling related to tumor progression.

The details of Ca^2+^ impacts on these signaling parameters will be developed in the next sections.

## 3. Role of Ca^2+^ Signaling in PDAC Pathophysiology

The role of the above transporters, together with the Ca^2+^ signaling they are involved in or directly trigger, is described in the context of the main pathophysiological processes of cancer and notably PDAC, such as autophagy, proliferation, apoptosis, migration, and invasion.

### 3.1. PDAC Cells and Autophagy

An abnormally high level of autophagy has been reported in PDAC cells. This can be correlated to a nuclear translocation increase of microphthalmia transcription factors [33]. Therefore, it has been shown that the overexpression of the latter increases the expression of the autophagy-involved genes.

In addition, Unc-51 like kinase-1 (ULK1), a kinase involved in autophagy, dephosphorylated by the phosphatase PP2A-B55α seems to be highly common in PDAC cells [34]. Although these mechanisms are hardly understood in PDAC, this over-activation of autophagy seems to increase cell proliferation, invasion and metastasis in this cancer [35].

Although few studies have been done regarding the Ca^2+^ role in autophagy in PDAC. In many other models, the Ca^2+^ role in autophagy regulation has been shown, allowing scientists to make this hypothesis for PDAC. Furthermore, its role in autophagy is not precisely identified as some studies reveal an anti-autophagic role and others, a pro-autophagic role [36].

In addition, the AMP-activated protein kinase (AMPK), an autophagy inductor is regulated by the AMP/ATP ratio, which depends on cellular metabolism but may also be activated by the Ca^2+^/Calmodulin (CaM) Dependent Protein Kinase Kinase β (CaMKKβ) [37]. This kinase is activated by the intracellular Ca^2+^ increase.

### 3.2. PDAC Cells and Proliferation

As in all types of cancer, PDAC cell proliferation is one of the main mechanisms, which explains its aggressiveness. In 90% of PDAC cells, the proto-oncogene K-RAS is mutated, causing its constitutive activation. This mutation is associated with a high Ca^2+^ concentration that favors signaling pathways involved in growth, invasion and/or metastasis [38].

A store operated Ca^2+^ entry (SOCE) mechanism could explain this concentration increase. Once the ER Ca^2+^ stores are empty, Ca^2+^ Release Activated Channels (CRACs) are open, allowing the cell to self-re-load with Ca^2+^. These channels are composed of ORAI1 channels and STIM1 sensor.

Therefore, the SOCE mechanism interferes with Akt/mTor [39] and NF-κB pathways interfering with proliferation [40]: following the Ca^2+^ entry in the cell, proteins, such as the Ca^2+^/CaMKK β would be activated, inducing nuclear translocation of Akt/mTor or nuclear factor of activated T-cells (NFAT).

### 3.3. K-RAS, Calmodulin and Proliferation

PDACs present varied clinical, pathological and genomic profiles [41], which are partially explained by the fact that wild-type PDAC KRAS can be divided into three categories: (i) PDAC with MAPK pathway over-activation often associated with BRAF dysfunction, (ii) PDAC with microsatellite instability (MSI) and (iii) PDAC with the kinase resulting from the gene fusion [42].

While the development of therapeutic molecules against KRAS-mutated PDAC remains a major challenge, therapies targeting wild-type KRAS are under development. Dabrafenib, a BRAF inhibitor, shows partial efficacy against BRAF-mutated PDAC [43]. In addition, pembrolizumab, an immune checkpoint inhibitor also called anti-PD1, is effective in some cases of MSI-involved PDAC [44]. Finally, kinase inhibitors have been shown to be effective against PDAC with the kinase resulting from the gene fusion [45].

CaM, a small Ca^2+^ binding protein [39], can induce differentiation [40], proliferation [46], survival and motility [47] by cooperating with the CaM-binding protein. Recent studies highlight that the Ca^2+^/CaM complex controls the KRas4B pathway due to the K-RAS mutation present in 90% of human PDAC cases [48]. This fact assumes the implication of the Ca^2+^/CaM complex, explaining the proliferation increase.

It is important to emphasize that MAPK pathways are often impaired in PDAC. In fact, these pathways are highly affected by mutations. The overactivation of the MAPK pathway induces the overexpression of p21cip1, a cyclin dependent kinase 1 (CDK1) inhibitor, leading to a cell cycle arrest. In contrast, a transient activation seems to induce cell proliferation [38,39,40,41].

Furthermore, the binding of CaM to K-RAS can decrease the activity of the MAPK pathway, preventing the anti-proliferation effect due to the overactivation [49] and the increase in the activity of the PI3K/Akt pathway [50]. Therefore, CaM slows down the overactivation of extracellular signal-regulated kinases 1/2 (ERK 1/2) and favors proliferation via the PI3K/Akt pathway.

To favor the PI3K/Akt pathway, the oncogene KRas4B binds to the CaM with its hypervariable C-term region [51]. Both protein will bind to PI3K to induce an increase in Akt activity [50], leading to several biological effects such as an increase in proliferation or cell migration through the matrix metalloproteinase-2 (MMP-2) expression [52].

Moreover, Ca^2+^ can cause activation of different ion channels like potassium or chloride channels. The Ca^2+^-activated potassium (K^+^) channel K_Ca_3.1 is notably overexpressed in PDAC inducing K^+^ efflux. This process promotes the proliferation of PDAC cells [53]. It has been also shown that K_Ca_3.1 inhibition by Triarylmethane-34 decreases tumor growth [54].

Many proteins related to Ca^2+^ are overexpressed in PDAC such as S100 proteins (particularly S100A14 and S100P) or PKM2 [55], a glycolytic enzyme that provides ATP to PMCA, a calcium exporter that regulates cancer cell homeostasis [26].

### 3.4. PDAC Cells and Apoptosis

Beyond its physiological role in cells, apoptosis is impaired in many cancers [6]. Resistance to the apoptosis in cancer cells can cause therapeutic failures and cancer recurrences. Ca^2+^ plays a central role in apoptosis, particularly in the intrinsic pathway, which could participate in the apoptosis-resistance of PDAC cells.

It was shown that chemotherapy can generate apoptosis resistance in PDAC by increasing SOCs expression. Indeed, the treatment with 5-fluorouracil (5-FU) or gemcitabine increases ORAI1 and STIM1 expression, favoring SOCE in PDAC [25]. When siORAI1 or siSTIM1 is used, only cells treated by the chemotherapy have their sensitivity to the apoptosis increased. Unfortunately, mechanisms involved in this process are poorly known but future studies could investigate these anti-apoptotic roles of ORAI1 and STIM1 in pancreatic adenocarcinoma in depth. Proteins regulated by Ca^2+^ can also participate in the chemoresistance. Interestingly, S100A4, overexpressed in PDAC, is involved in chemoresistance by decreasing the proapoptotic BCL2 interacting protein 3 (BNIP3) expression and desensitizing cells to apoptosis [56].

It is known that a Ca^2+^ overload induces apoptosis. PMCA4 seems to be overexpressed in PDAC and thus involved in the apoptosis resistance by “Ca^2+^ clearance” mechanisms, delaying the Ca^2+^ overload [17]. This protein, and PMCAs in general, are positively regulated by phosphofructokinase fructose biphosphate 3 (PFKFB3) that is responsible for low intracellular Ca^2+^ levels, and is also overexpressed in PDAC [57]. Tissue transglutaminase 2 (TG2), highly expressed in PDAC, can be also activated by Ca^2+^ overload. Its activation induces apoptosis-inducing factor (AIF) release by mitochondria. These factors will be translocated into the nucleus and will induce fragmentation, causing apoptosis [58]. Targeting PCMA4, tissue TG2 or PFKFB3 could restore treatment sensibility to PDAC cells by increasing apoptosis.

Many links among Ca^2+^ signaling, mitochondrial fission and apoptosis were already identified. Recently, it was shown that the single-pass membrane protein with the aspartate (D)-rich tail 1 (SMTD1), a regulator subunit of the MCU complex, allows Ca^2+^ entry in the mitochondrial matrix during apoptosis [14]. In fact, the SMTD1 could stimulate mitochondrial fission and intrinsic apoptosis in PDAC.

The tumor micro-environment can promote tumorigenesis. For example, gastrin, a hormone secreted by the stomach, the duodenum and the pancreas, could play an anti-apoptotic role in PDAC [59]. This hormone seems to be able to bind to the cholecystokinin B receptor (CCKBR), a receptor mainly localized in the central nervous system, and induces the Akt pathway by Ca^2+^-dependent pathways (Ca^2+^ mobilization), but protein kinase C (PKC)-independent. Once activated, Akt will be able to inactivate BCL2 associated agonist of cell death (BAD) and forkhead box O3 (FOXO3), (a pro-apoptotic transcription factor) via phosphorylation, substantially slowing down apoptosis mechanisms.

Finally, the role of Ca^2+^-signaling and Ca^2+^-dependent pathways cannot be underestimated in the light of PDAC cell survival and is summarized in the Figure 2 below.

### 3.5. Role of Ca^2+^ Signaling in PDAC Cell Motility

The level of aggression of PDAC goes far beyond cell survival and basal apoptosis resistance where PDAC cells’ ability to migrate and metastasize are the key points in poor prognosis and cancer aggressivity. We focus on the role of the Ca^2+^ -dependent mechanisms and Ca^2+^-transporters in PDAC cell mobility and metastasis formation.

TRP channels are involved in Ca^2+^ homeostasis, inflammation or cell motility, making them a major target in the development of cancers [60]. TRPM2 overexpression has a role in the migration in PANC-1 cells, given that it induces filopodia formation by regulation of parvin gamma and sirtuin-2 genes [61]. Another Ca^2+^ channel, TRPV6 is also overexpressed and has a role in migration of PC cells [62]. Furthermore, TRPC1 channel, a non-specific Ca^2+^ channel, is required for the growth factor-driven directional migration. Indeed, this channel will produce an influx of Ca^2+^ that will activate chloride channels and thus induce migration [60].

Acid-sensitive ion channels (ASIC) are activated by the microenvironment acidity [63]. They are overexpressed in PDAC cells, such as PANC-1 and BxPC-3 cells. Their inhibition leads to the reduced migration induced by the acidity of the PDAC cells [64]. These channels have already been shown to be permeable to Ca^2+^ [63]. The acidity of the microenvironment contributes to an increase in intracellular Ca^2+^ level by activating the ASIC-1 and ASIC-3 channels in PDAC cells (Figure 3). The Ca^2+^ increase contributes to the migration of PDAC cells and endows them with the epithelial-mesenchymal transition (EMT) capacity in an acidic microenvironment. The entry of Ca^2+^ contributes to the activation of the Ras homolog family member A protein (RhoA), which is essential for this EMT process [64].

A recent study demonstrated that inositol triphosphate receptor (IP3R) channels moved in PANC-1 cells from cell-cell contacts to the leading edges of migrating cells after TGF-β1 stimulation [13]. The IP3R channels, thus, move towards the lamellipods and the filopodia at the focal adhesion level (Figure 3). This delocalization and accumulation of IP3R channels at the migration front increase the Ca^2+^ sensitivity in this region and promote cell migration. This results in a local increase in Ca^2+^ and paxillin, a protein involved in focal adhesions. Interestingly, they also demonstrated that the change of localization of IP3R channels is accompanied by the accumulation of ER-PR junctions and STIM proteins. Inhibitors of IP3R and STIM channels help to decrease PDAC cell migration. Therefore, a formation of a structural and functional polarity in migrating PDAC cells was observed. The disruption of this polarity could be a way to fight cancer development [13].

It is known that the K^+^ and Cl^−^ channels are also key elements for the regulation of cell volume, a necessary process for cell migration. K_Ca_3.1 seems to play an important role in PDAC development [65]. A recent study has demonstrated for the first time the involvement of this channel in the migration of PDAC cells [53]. They demonstrated that the K_Ca_3.1 channel was overexpressed in PDAC cells, and its inhibition resulted in the loss of cell migration. Moreover, a K^+^ current was activated by Ca^2+^. Being located at the rear pole of migrating cells (Figure 3), the activation of these channels by Ca^2+^ would allow the efflux of ions and water, contributing to the decrease in cell volume and the retraction of the rear pole of the cell [53].

As concerns the chloride channels, the inhibition of the anoctamin-1 (ANO1) channel (TMEM16A), a Ca^2+^-activated Cl^−^ channel, decreases the migration of AsPC-1 and BxPC-3 cells [66]. Another study showed that in these cells, inhibition of ANO1 by siRNA decreased migration, even after epidermal growth factor (EGF) stimulation. At the same time, in control cells, EGF induced cell migration via Ca^2+^ signaling as well. Inhibition of ANO1 results in a decrease in Ca^2+^ entry induced by EGF. In fact, ANO1 facilitates EGF-induced Ca^2+^ release from intracellular stores and thus promotes migration [67].

### 3.6. Role of Ca^2+^ Signaling in PDAC Cell Invasion

CRAC seems to affect the development and growth of many cancers [68]. The over-expression of STIM1 results in metastases. Histological studies of cancer cells show that patients with higher STIM1 expression have bigger tumors with higher nerve invasion [69]. STIM1 also regulates the oncogenic phenotype of PDAC. In addition, the KD of STIM1 has reduced the invasiveness by approximately 49%. STIM1 has a central role in the invasion of PDAC [69]. Subsequently, a link between the hypoxia-induced factor 1-alpha (HIF-1α) and the STIM protein was identified in PDAC. In fact, in PDAC, HIF-1α is upregulated as compared to its expression in normal paired tissues. The overexpression of HIF-1α induces an increase of STIM1-α in the ER. In fact, HIF-1α binds to the human EGF receptor 2/3 (HRE2/3) sequence which will induce the overexpression of the *stim1* gene. STIM1 is therefore overexpressed in a hypoxic microenvironment and may contribute to the PDAC cell invasion, and thus, cancer progression [69].

The ASIC-1 and ASIC-3 channels contribute to the invasion of PDAC cells (Figure 3). In the PANC-1 and BxPC-3 lines, the use of both siASIC-1 and siASIC-3 decreases invasion. In the acid microenvironment, the activation of these channels allows the invasion of PDAC cells through the activation of the RhoA which promotes the EMT phenotype [64].

In the same way as migration, K_Ca_3.1 channels are also involved in the invasion of PDAC cells (Figure 3). Overexpression in the MiaPaCa-2 and BxPC-3 lines, their inhibition leads to a decrease in invasion. While acting through an efflux of K^+^ activated by Ca^2+^, they contribute to the regulation of cell volume [53].

Proteins of the S100 family are small Ca^2+^-binding activated proteins [70]. They are involved in cell invasion through the Ca^2+^ signaling and the direct interaction with the cytoskeleton components [70]. Their expression can be modified in pathology like PDAC to induce invasion. The S100A4 is especially overexpressed in PDAC mesenchymal cells, and represents an EMT marker [58,59,60]. The EMT is a specific process of the tissue invasion by cancer cells issue from carcinomas. A S100A4 KD, having no effect on cell viability leads to a dramatic decrease of cell invasion capacity both in vitro and in vivo [71]. Among the S100 proteins, S100A14 is a mesenchymal-epithelial transition (MET) marker due to its overexpression in epithelial cells [72]. A S100A14 KD has induced the appearance of EMT morphologic characteristics, with the increase of the S100A4 expression, following by the increase of cell invasion. The S100A14 inhibition in turn increased cell invasion capacity [71]. The overexpression of S100A4 protein, observed in 60% of PDAC patients [72], is closely linked to the high invasiveness and ability to metastasize in PDAC, and was associated with a poor diagnosis [72].

### 3.7. Role of Ca^2+^ Signaling in PDAC Metastasis Formation

Ca^2+^ signaling can also be indirectly responsible for the development of metastatic phenotype of PDAC, for example when a K^+^ Ca^2+^-activated channel subfamily N member 4 (KCNN4), interacts with the gamma-aminobutyric acid (GABA) type A receptor subunit Pi (GABRP) [73].

Indeed, some studies show that neurotransmitters play a role in tumor progression by stimulating the migration and invasion capabilities of cancer cells [74]. The microenvironment of the PDAC tumor is rich in neurotrophic factors, derived from glial cells, that may increase the invasion capabilities of PDAC cells [75]. GABA is the main inhibitory neurotransmitter of the central nervous system and is found in different endocrine organs. The GABA has three different receptors, particularly, the gamma-aminobutyric acid A receptor (GABR), composed of five subunits including GABRP (Figure 3).

The impact of GABRP on tumor progression has already been demonstrated in breast and ovarian cancers. GABRP level is also increased during the malignant transformation of PDAC correlating with an increase in the metastasis process. In addition, the inhibition of GABRP expression suppresses pulmonary metastases caused by PDAC. GABRP is able to interact with the KCNN4 receptor (also called K_Ca_3.1). Indeed, the KCNN4 channel plays a role in the intracellular Ca^2+^ level since its inhibition induces a reduction in the intracellular Ca^2+^ level. The activation of the KCNN4 channel increases the level of C-X-C motif chemokine ligand 5 (CXCL5) and Chemokine (C-C motif) ligand 20 (CCL20) while its inhibition leads to the reduction in these levels. Therefore, GABRP recruits the KCNN4 channel at the level of the lipid rafts, and activates them, allowing an increase in the intracellular Ca^2+^ concentration. Intracellular Ca^2+^ augmentation induces activation of the NF-κB transcription factor for the expression of CXCL5 and CCL2. The expression of CXCL5 and CCL2 induces the recruitment of macrophages at the tumor site, which has the consequence of increasing the metastasis potential of PDAC cells [73].

Several studies have already shown the involvement of CRAC channel (as in many cancers such as breast cancer, prostate cancer, glioblastoma, etc.), in metastatic process, as well as in the resistance to apoptosis of PDAC cells [25]. Overexpression of STIM1 reveals bad diagnosis, which results in the presence of metastases, especially lymph and hepatic metastasis. Moreover, the overexpression of HIF-1α is associated with the increase of metastasis [69].

A study [68] has shown the implication of ORAI1/STIM1 interaction in the metastasis process of PDAC via using of a small molecule inhibitor of the CRACs (RP4010) that significantly decreased the number of colony formation. The reduction in cell proliferation induced by the RP4010 inhibitor induces a reduction in the Akt/mTOR pathway, including the p70-S6 (S6K) enzyme. Therefore, the involvement of the CRAC ORAI1/STIM1 channel in the metastasis process for PDAC is likely to be evident. Evidence of its involvement has been shown in prostate, breast, glioblastoma and cervical cancers, as well as in the resistance to apoptosis in PDAC. Taken together, these results suggest that the CRAC channel could be involved in the PDAC metastasis process.

The TRP channels allow metastatic cancer cells to respond to the environmental signals, which is important because metastatic cells interact with a wide variety of environmental stimuli. In the metastatic process of PDAC, an overexpression of TRP channels with the permeability for Ca^2+^, including the TRPM2 channel was observed. An overexpression of TRPM2 was also associated with the metastatic process [61].

Finally, it has been shown that KD of ASIC-1 and ASIC-3 channels in mice xenograft model of BxPC-3 cells contributes to the reduction of hepatic and pulmonary metastases. Being involved in the mobility and metastasis processes of PDAC cells, the ASIC-1 and ASIC-3 channels could be interesting therapeutic targets [64].

In conclusion, the data demonstrate that Ca^2+^ signaling via Ca^2+^ transporter proteins is very important player in PDAC cell aggressivity and ability to invade. Their cell localization and intracellular signaling is depicted in the Figure 3. 

## 4. Ca^2+^ Signaling in PDAC Treatment

As it was shown above, Ca^2+^ transporters are involved in many oncogenic mechanisms in PDAC: proliferation, apoptosis, migration, invasion and metastasis. This would explain the emergence of anti-PDAC therapies, which target Ca^2+^-signaling players (Figure 4).

### 4.1. Are Ca^2+^ Transporters Potential Targets for PDAC Treatment?

In the end of the 20th century, Ca^2+^ channel blockers, such as Verapamil, were discovered and shown to inhibit tumor progression of the chemotherapy-resistant PC side population cells [76]. From that point, a multitude of treatments has emerged that seem to be efficient in curing PDAC.

#### 4.1.1. Impact on Proliferation

It was found that when a component of the multimeric structure of ORAI1 is KD in MiaPaCa-2 cells, it reduces tumor proliferation. Moreover, a compound RP4010 was found as a CRAC inhibitor. Protein studies showed the action of RP4010 in such signaling pathways as Akt/mTOR, NFAT and NF-κB, and the consequences of RP4010 on the expression of proteins involved in these pathways. The combination with other compounds are considered as usual standard cancer treatments: Gemcitabine (it incorporates into DNA by mimicking cytidine and it inhibits nucleotide synthesis, which leads to the cell death) and nab-paclitaxel, leading to the accentuation of the effect by decreasing the cell growth [68]. Furthermore, the treatment combination of RP4010, gemcitabine and nab-paclitaxel was shown to down-regulate the pro-survival Ca^2+^ signaling pathways indicated above, but more work needs to be done.

#### 4.1.2. Impact on Migration and Invasion

Further, PDAC is characterized by the formation of a dense fibrotic stroma (desmoplasia), primarily formed by activated pancreatic stem cells (PSCs). Desmoplasia contributes to high pancreatic tissue pressure, which activates PSCs. The migration of PSCs is enhanced in the pressurized PSCs, however, the KD of TRPC1 led to the attenuated phenotype [12]. A recent study [13] also showed that the migration of PDAC cells was strongly suppressed by selective inhibition of IP3Rs and SOCE, indicating that these mechanisms are functionally required for migration.

Concerning migration, there is also an involvement of another channel called TRPM7. Indeed, the downregulation of TRPM7 by siRNA dramatically inhibited both intracellular Mg^2+^ fluorescence (decrease in Mg^2+^ concentration) and cell migration without affecting cell proliferation, suggesting that TRPM7 contributes to Mg^2+^ entry and cell migration. In addition, TRPM7 expression was shown to be inversely related to the patient survival [10].

Sometimes there can be a cooperation between two channels observed [11]: the K_Ca_3.1 channel downregulation or KD prevents the stimulation of PSCs migration and chemotaxis by reducing the intracellular Ca^2+^ concentration [Ca^2+^]_i_ and calpain activity. K_Ca_3.1 channels functionally cooperate with TRPC3 channels (cationic channels) that are upregulated in PDAC stroma. KD of TRPC3 channels prevents the impact of K_Ca_3.1 channels on PSCs migration, and therefore it would probably disadvantage the progression of PDAC.

#### 4.1.3. Impact on Some Other Mechanisms

Another interesting example is the PMCA, which is critical for maintaining a low [Ca^2+^]i, and which utilizes glycolytically derived ATP in PDAC. The inhibition of glycolysis in PDAC cell lines results in ATP depletion, PMCA inhibition and an irreversible [Ca^2+^]i overload with a significant decrease in both glycolytic flux and proliferation rate [15]. It is possible to inhibit glycolysis with the SHK which is an inhibitor of the PKM2. This enzyme contributes to ATP production that can be used by the PMCA, and thus, its inhibition permits to reduce PDAC cells proliferation, migration and to increase cell death [26].

The combination of fendiline (a L-type Ca^2+^ channel blocker) with other inhibitors (e.g., Gemcitabine) shows enhanced anti-tumor activity in PANC-1, MiaPaCa2 and CD18/HPAF PDAC cells, being evident from the reduced viability, migration, anchorage-independent growth (AIG) and self-renewal [77].

Finally, mahanine, a carbazole alkaloid extracted from a plant called *Murraya Koenigii*, can impair heat shock chaperone protein 90 (HSP90) activity involved in protein folding induced by reactive oxygen species (ROS) in PDAC [18]. Therefore, mahanine can induce apoptosis in PDAC following ER stress using Ca^2+^ pathways activation, such as ROS-induced Ca^2+^ leakage from the ER or even by activating UPR pathways.

Taken together, these results demonstrate that Ca^2+^ transporters can be interesting targets in the aim to treat PDAC. Their targeting can disrupt tumor progression by affecting oncogenic mechanisms like proliferation and migration because of the impact on the metabolic pathways impaired by the tumorigenesis process.

### 4.2. What Is Their Place in Drug Resistance?

Despite the huge advancements in the discoveries of major treatments for the disease, the most of patients are developing resistance to the previously identified treatments. The Gemcitabine is usually used as a combination with other molecules like the Nab-paclitaxel [78]. Hence, a multitude of scientific works showed how cancer cells resist and escape previously discovered treatments.

The combination of Gemcitabine/5-Fluorouracil blocks SOC channels in PANC-1, an action that cancels resistance to the apoptosis provided by these channels. The KD experiments with siORAI1 and siSTIM1 confirmed that ORAI1 or STIM1 decreased cell death [25].

Gemcitabine can be also combined with an inhibitor of the fatty acid synthase called orlistat. Indeed, this enzyme becomes overexpressed while disease progresses in spontaneous PC mouse models. In addition, there is a positive correlation between its expression and patient survival associated with a poor Gemcitabine responsiveness in cell lines. The combined treatment significantly diminished cell stemness, in part, due to the induction of ER stress which resulted in apoptosis [79].

Further, the Gemcitabine resistance was studied on a genetic level induced by the long non-coding RNA plasmacytoma variant translocation 1 (PVT1) gene. TUNEL assays showed a decrease in the number of apoptotic cells in PANC-1- and ASPC-1-Gemcitabine resistant cell lines. PVT1 action has been shown to be associated with the activation of both Wnt/β-catenin signaling pathway and autophagy, rendering thus a chemoresistance. The resistance is manifested by a decrease of miR-619-5p that will upregulate the autophagosome activity and thus induce Gemcitabine resistance [80].

Another Gemcitabine resistant cell line was established, which permitted to identify the action of ribonucleoside-diphosphate reductase large subunit (RRM1) on the transcriptomic level. That process resulted in cell survival and another mechanism concerning STIM1 on an epigenetic level has been identified. In fact, Ca^2+^ signaling is induced via SOC channels following ER stress via STIM1 action. This Ca^2+^ increase will lead to a differential expression of some proteins via inhibiting the activating transcription factor 4 (ATF4) transcription and increase in NFAT activity [81].

In addition, one more resistance has been manifested passing via the energy metabolism. Indeed, the transition to a glycolytic metabolism allows PMCA to maintain a homeostatic balance even in the state of hypoxia by preventing cytotoxicity in cancer cells due to an excess of Ca^2+^ [82]. Indeed, the role of hypoxia and hypoxia-related mechanisms may be very important in the metabolism of PDAC as in some cancer types, such as clear cell renal cell carcinoma, where hypoxia displays some interactions with the calcium-related pathways [83]. Further work in this field needs to be done as hypoxia-related mechanisms may control the metabolism of some cancer types such as clear cell renal cell carcinoma [84]. For example, Tian and Wang [85] have identified in 65 patients having PDAC some upregulated differentially expressed genes. Among those, genes regulated in response to hypoxia, calcium signaling and negative regulation of catecholamine, were predominant. On the other hand, understanding PDAC metabolomics is crucial since targeted metabolomics approaches are versatile in quantifying multiple metabolite levels in a single platform, and thus, enable the characterization of multiple metabolite alterations. A new metabolomics approach has been elaborated recently while using a liquid chromatography-coupled tandem mass spectrometry. This approach could characterize the hypoxia-induced metabolomics alterations using pancreatic cancer cell lines cultured in normoxic and hypoxic conditions [86]. It provides new opportunities for to the hypoxia-calcium signaling studies in the foreseeable future.

Eventually Ca^2+^ channels seem to be involved in drug resistance (Figure 4). Efforts to combine classical drugs with others types of drugs have been made and the results are encouraging as they principally managed to increase cell death.

## 5. General Conclusions and Perspectives

Throughout this review, the importance of Ca^2+^ channels in both survival and apoptotic phenomena in PDAC cancer cells has been investigated. The transporters involved in PDAC can be channels (Ca^2+^-channels or cationic channels), pumps or even exchangers with different subcellular localizations: PM, mitochondria, ER or Golgi apparatus. The direct involvement of some Ca^2+^-dependent proteins like Ca^2+^/CaM-dependent protein kinase type II (CaMKII) has been reported [87]. The over-expression (e.g., *PMCA4* [88]) or the under-expression (e.g., Single-Pass Membrane Protein With Aspartate Rich Tail 1 (SMDT1) [14]) of some Ca^2+^ transporters in PDAC increases cancer cell survival and proliferation. This observation indicates the importance of Ca^2+^ signaling pathways whereas their alteration can promote cell migration and invasion. PDAC cells also present an increase in their autophagic activity, which can contribute to a survival mechanism.

Therefore, the implication of Ca^2+^ remains to be understood. PDAC is still considered as the life-threatening disease, and despite a lot of studies, no effective treatment exists so far. Standard care treatments affect only the tumor in general, and diagnostic markers still lack sensitivity to adequately diagnose PDAC at early stages. Some genes related to the high risk of PDAC, such as breast cancer 2 gene BRCA2 or PALB2, are known but only 10 to 20% of PDAC can be diagnosed using them [89]. Surgical resection is the only chance for curative treatment of PDAC and it mostly depends on when the decision to do surgery is taken, hence, highlighting the importance of early detection [89]. In addition, drug resistance often appears in this cancer and is a reason to move more and more towards combined treatments. Therefore, targeting Ca^2+^ transporters may represent an important perspective in the foreseeable future.

## Figures and Tables

**Figure 1 cancers-13-03085-f001:**
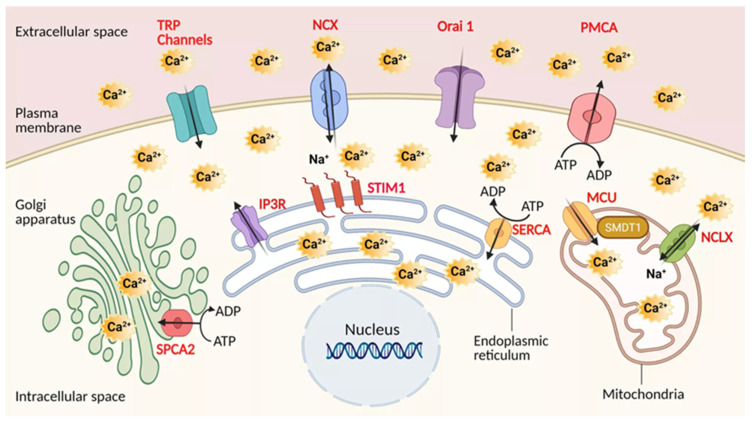
Calcium transporters in Pancreatic ductal adenocarcinoma (PDAC). Channels, pumps, exchangers are depicted in regards to their localization. Plasma membrane: Transient receptor potential channels (TRPs), The sodium-calcium exchanger (NCX), Calcium release-activated calcium channel protein 1 (Orai1), Plasma membrane Ca^2+^ ATPase pump (PMCA). Mitochondria: Mitochondrial calcium uniporter (MCU), Mitochondrial sodium-calcium exchanger (NCLX). Endoplasmic reticulum: Sarco-Endoplasmic Reticulum Calcium ATPase pump (SERCA), Inositol trisphosphate receptor (IP3R). Golgi apparatus: secretory pathway calcium ATPase pump (SPCA). Regulatory elements: Stromal interaction molecule 1 (STIM1), Single-Pass Membrane Protein with Aspartate Rich Tail 1 (SMDT1) MCU subunit. Created using BioRender.com source, accessed on 8 March 2021.

**Figure 2 cancers-13-03085-f002:**
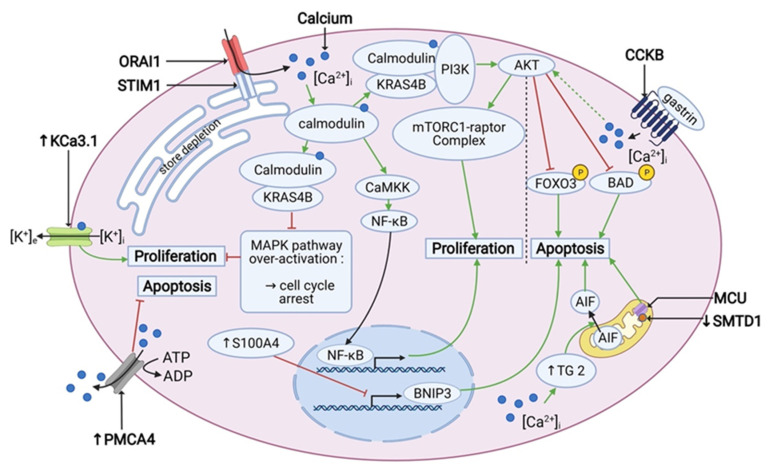
Calcium signaling is involved in the cell proliferation and apoptosis in PDAC. PDACs overexpress PMCA4 and TG2, which have an anti-apoptotic and pro-apoptotic effect, respectively. Also, SMTD1 and BNIP3, which have a pro-apoptotic effect, are under-expressed in PDAC. Under-expression of BNIP3 is driven by over-expression of S100A4. Gastrin has an anti-apoptotic effect via an inhibitory phosphorylation of FOXO3 and BAD by AKT. SOCE activates both the PI3K/AKT/mTOR pathway and the nuclear translocation of NF-κB, which promotes proliferation. In addition, it prevents the overactivation of the MAPK pathway, which can lead to the cell cycle arrest. Also KCa3.1 channel is overexpressed and enhances proliferation. “↑” stands for overexpression and “↓” stands for under-expression. Created using BioRender.com source, accessed on 8 March 2021.

**Figure 3 cancers-13-03085-f003:**
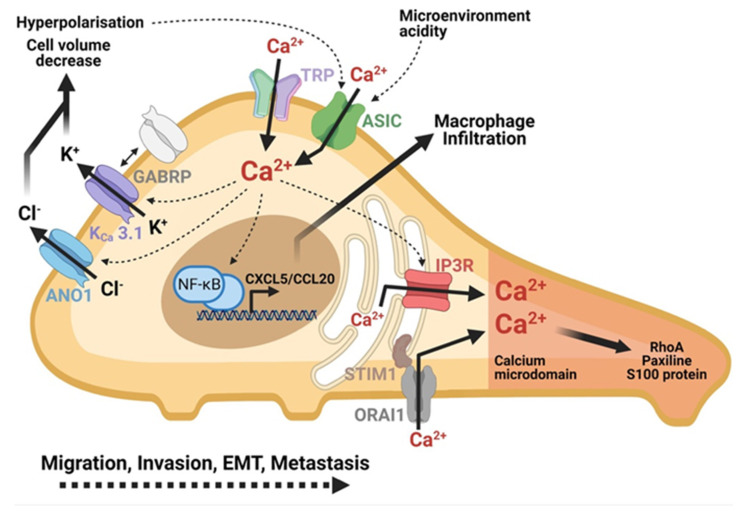
Role of calcium in PDAC cells mobility and metastasis formation. Calcium plays a central role in migration, invasion and metastasis of PDAC cells. The microenvironment activates the ASIC and TRP channels which contribute to the increase in the intracellular calcium. IP3R channels and SOC-type channels (ORAI1) amplify this increase of intracellular calcium. The formation of a calcium microdomain at the migration front leads to the activation of proteins essential to cell mobility: RhoA, Paxillin, S100 proteins. Calcium entry also indirectly activates the transcription of chemokine-like genes to promote inflammation and invasion. Calcium activates chlorine (ANO1) and potassium (KCa 3.1) channels, which play a role in the regulation of cell volume at the rear pole of the cell in the hyperpolarization of the cell. Created using BioRender.com source, accessed on 8 March 2021.

**Figure 4 cancers-13-03085-f004:**
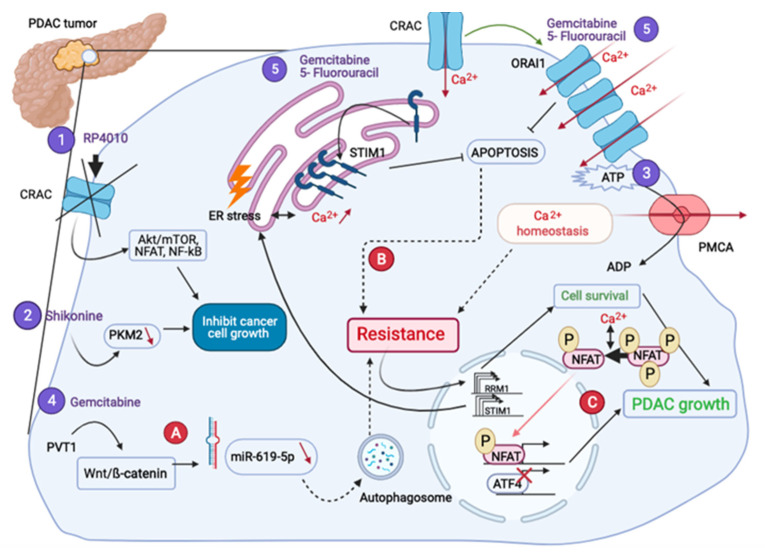
PDAC treatments and different therapeutic resistances. RP4010 (1) and Shikonine (2) treatments inhibit cell growth. The inhibition of the ATP used by the PMCA (3) decreases the proliferation rate. Gemcitabine combined treatments (4–5) leading to the chemoresistance, pass through the autophagic activity (A), and a decrease of the apoptosis (B), respectively. The resistance to Gemcitabine also leads both to the overexpression of RRM1 inducing cell survival and STIM1, activating the NFAT-dependent transcriptional program (C).

**Table 1 cancers-13-03085-t001:** Expression, localization, and the potential role of calcium transporters in pancreatic ductal adenocarcinoma (PDAC).

Calcium Transporters [RT1]	Calcium Transporter Type	Localization	Alteration	Involved Cancers	Ref
**Channels**					
TRPM7 (transient receptor potential melastatin related 7)	Cationic (Ca^2+^) channels	Plasma membrane	Migration and/or invasion of cancer cells: metastasis formation	PDACPancreatic ductal adeno carcinoma, epidermal, neuroblastoma, glioblastoma, breast cancer, nasopharynx cancer, lung cancer, prostate cancer	[10]
TRPC3 (Transient Receptor Potential Cation Channel Subfamily C Member)	Cationic (Ca^2+^) channels	Plasma membrane	Proliferation and metastasis formation	Pancreatic ductal adenocarcinoma	[11]
TRPC1 (Transient Receptor Potential Cation Channel Subfamily C Member)	Mechanosensitive channel	Plasma membrane	Altered TRPC1 with tissue pressure	Pancreatic ductal adenocarcinoma	[12]
Orai1 (SOCC: store-operated Ca^2+^ Channel)	Calcium channel	Plasma membrane	Migration and/or invasion of cancer cells: metastasis formation	Pancreatic ductal adenocarcinoma	[13]
IP3R (Inositol trisphosphate receptor)	Receptor channel	Endoplasmic reticulum membrane	Migration and/or invasion of cancer cells: metastasis formation	Pancreatic ductal adenocarcinoma	[13]
**Pumps & Transporters**					
MCU (mitochondrial calcium uniporter complex)	Transmembrane receptor	Membrane of mitochondria	Altered mitochondrial dynamics	Pancreatic ductal adenocarcinoma	[14]
SMDT1 (Single-Pass Membrane Protein with Aspartate Rich Tail 1)	Essential regulatory subunit of the MCU				
PMCA (plasma membrane Ca^2+^ ATPase)	Transport protein of Calcium (pump)	Plasma membrane	High calcium efflux	Pancreatic ductal adenocarcinoma	[15]
PMCA1 (plasma membrane Ca^2+^ ATPase isoform 1)	Transport protein of Calcium	Plasma membrane	Upregulated	Pancreatic ductal adenocarcinoma	[16]
PMCA4 (plasma membrane Ca^2+^ ATPase isoform 4)	Transport protein of Calcium	Plasma membrane	Cell migration and apoptotic resistance	Pancreatic ductal adenocarcinoma	[17]
SERCA2 (sarco/endoplasmic reticulum Ca^2+^-ATPase)	Calcium ATPase-type Pump-ATPase	Endoplasmic reticulum membrane	Upregulated	Pancreatic ductal adenocarcinoma	[18]
SLC24A2 (NCX1,2,3) (Solute Carrier Family 24 Member 2)	Sodium/Potassium/Calcium Exchanger	Plasma membrane	Mutation in SLC24A2 gene decreases the stability of SLC24A2, tumor progression	Pancreatic ductal adenocarcinoma	[19]

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
