# Peer review of "Ca2+ Signaling and Its Potential Targeting in Pancreatic Ductal Carcinoma"

_cancers, 2021, doi:10.3390/cancers13123085_

Round 1
Reviewer 1 Report
Bettaieb et al. reviewed the role of Ca²⁺ signaling and its potential targeting in pancreatic ductal adenocarcinoma and in drug resistance. The topic is very interesting and the manuscript is well-written. The literature's review is adequate and the conclusions are supported by published evidences.
Author Response
We would like to thank the Reviewer 1 for the objective reviewing of the manuscript.
We did our best to choose the right topic and to write our manuscript adequately.
Thank you.
Reviewer 2 Report
This is a solid paper and a timely contribution to the field.
The authors should be congratulated for this excellent work and also for providing paradigmatic figures, very useful for the readers.
Some notes for improving the manuscript:
1) the paragraph on K-RAS, calmodulin and proliferation is very interesting. Please provide more discussion and more text, with in-depth analysis, on the differences in terms of calciium-related mechanisms, in pancreatic ductal adenocarcinoma KRAS mutated vs. KRAS wild type. KRAS wild-type pancreatic ductal adenocarcinoma represents about 10% of pancreas cancer, but there are many potential targets for molecular-based therapies in this setting (suggested references PMID: 32350089 ; 30836094 ; 32933947 ; 33115526 ), thus it is of great interest a focused analysis on this peculiar topic.
2) Are there any differences on calcium-based mechanisms among different histotypes of exocrine cancers? (e.g.: pancreatic ductal adenocarcinoma: conventional vs. variants; pancreatic ductal adenocarcinoma vs. acinar cell carcinoma; pancreatic ductal adenocarcinoma vs. solid pseudopapillary neoplasms). Please expand this part also indicating future directions for Calcium-related researches in this field based on distinguished histology.
3) hypoxia-related mechanisms are important in the metabolism of some cancer types, such as clear cell renal cell carcinoma, and display some interactions with calcium-related pathways. Any of these associations in pancreatic cancer? In any case, please describe the importance of metabolomics as another important -omic science in this field.
Minor:
different size of letters in the text of the introduction. please use the same font and size.
Author Response
We would like to thank the Reviewer 2 for the objective reviewing of our manuscript.
We are happy to know that the Reviewer 2 finds our work solid and a timely contribution to the field.
We have tried our best to provide paradigmatic figures, very useful for the readers.
Please find below our response to your comments:
- The Reviewer 2 suggests providing more text with in-depth analysis, on the differences in terms of calcium-related mechanisms, in pancreatic ductal adenocarcinoma KRAS mutated vs. KRAS wild type. Moreover, the Reviewer suggests adding some references which we have integrated into the manuscript. All these references (PMID: 32350089 ; 30836094 ; 32933947 ; 33115526 ) were successfully added and the discussion on the differences in the terms of calcium-related mechanisms, in KRAS-mutated pancreatic ductal adenocarcinoma was provided in the Pages 9-10.
- The Reviewer 2 inquiries whether any differences on calcium-based mechanisms among different histotypes of exocrine cancers exist. In addition, even the Reviewer requests some indication future directions for Calcium-related researches in this field based on distinguished histology. Consequently, we have searched for the additional data on this subject and have incorporated a text and a discussion into the introduction section, Page 2.
- The Reviewer 2 suggests that hypoxia-related mechanisms are important in the metabolism of some cancer types, such as clear cell renal cell carcinoma, and display some interactions with calcium-related pathways. Indeed, though few data are available on this subject we have included some important works and discussed the perspectives in the text regarding the importance of metabolomics as another important -omic science in this field (Pages 17-18).
- The Reviewer 2 has noticed different size of letters in the text of the introduction. We have fixed this problem. Thank you.
Reviewer 3 Report
The manuscript is a review article demonstrating the role of calcium signaling as potential target in pancreatic ductal carcinoma. The authors have presented the discussion on various calcium channels, transporters, and pumps which are critical in pancreatic cancer. Besides the authors have also discussed about the role of calcium signaling in proliferation, motility, invasion, and death in pancreatic cancer. Finally, the authors highlight the significance of targeting calcium signaling in the treatment of pancreatic ductal carcinoma. Overall, the review is well organized, and presented. There are few minor suggestions.
- The authors have abruptly started the section about the information of various calcium channels, transporters, and pumps. As a suggestion, should start with short introduction.
- Table 1 can be shifted after the discussion on calcium transporters.
- Similarly, the role of calcium on various signaling parameters requires an introduction followed by description, which is missing.
- Section 4, describing the role of calcium signaling on mobility and metastasis can be merged with the section 3.
Author Response
We would like to thank the Reviewer 3 for the critical reviewing of our manuscript.
We are happy to know that the Reviewer 3 finds our work well organized, and presented.
Please find below our response to your comments:
- The Reviewer 3 has suggested a short introduction before the section Ca²⁺ transporters in PDAC). It was done and calcium transporters were introduced correspondingly (Page 3).
- The Reviewer 3 has suggested a shift in the Table 1 localization after the corresponding section on the calcium transporters. We have shifted it as suggested by the Reviewer 3 (Pages 7-8).
- The Reviewer 3 requests a brief introduction on the role of calcium in various signaling parameters. We have introduced this introduction where requested (Pages 3 and 6).
- The Reviewer 3 suggests merging sections 3 and 4 together. We do agree with the Reviewer 3 and have merged sections 3 and 4. Thank you.
Reviewer 4 Report
- We request that the language (not grammar) to be checked by a native/professional editor. We encountered multiple instances where the wording seemed inaccurate. For example, please correct the word ‘overlook’ (means fail to notice) into ‘overview’ and ‘signalization’ into ‘signaling’ and ‘actors’ into ‘factors’ in the following sentence from the abstract:
‘’This work provides a complete overlook at the Ca²⁺ signalization and its main actors in PDAC.’’
- We propose an alternative title which we think would be more scientifically meaningful: ’Calcium signaling and its downstream targets regulating pancreatic ductal adenocarcinoma’
- In section 2.1, the introduction on ion channels is not required. The audience are well aware of such trivial molecular biology terms.
- The table is not necessary since it is well explained in the manuscript. We request to omit it all together.
- We observed that in few sections, the authors cited the reference almost word-to word which might have the risk plagiarism. For example for ref. 15, the original authors stated in the abstract, ‘…the glycolytic dependence of the PMCA is a specific vulnerability of PDAC cells exhibiting the Warburg phenotype.’
While the authors of the current review wrote, ‘….the authors have shown that the glycolytic dependence of the PMCA is a specific vulnerability of PDAC cells presenting a Warburg phenotype.’
Ref 15: James, A.D. et al. The Plasma Membrane Calcium Pump in Pancreatic Cancer Cells Exhibiting the Warburg 739 Effect Relies on Glycolytic ATP. J Biol Chem 2015, 290, 24760–24771, doi:10.1074/jbc.M115.668707.
- We suggest the authors to include the full form of a molecule along the abbreviation when mentioned for the first time. For example, MCU, NCX.
- Figure 1 is redundant. It overlaps with Figure 2. We suggest to exclude it for the clarity.
Overall, the length of the article fits better for a book chapter than a short, concise review.
Most importantly, we have the impression that the review article represents significantly broad overview of calcium signalizing involved in cancer but not particularly in PDAC. Instead, it would be beneficial to the audience if there were a specific aim, for example Calcium signaling involved in the cell proliferation, apoptosis and/metastasis. In many cases, the article only suggested what pumps or channels could play a potential role in development of PDAC and urged the community to investigate them without showing that they have known role in other types of cancers. This makes the article weak in its presentation, as it does not provoke realistic ideas/suggestions nor it discusses the established findings.
Author Response
We would like to thank the Reviewer 4 for the critical assessment of our manuscript that has helped us to improve its quality.
Please find below our response to your comments:
- The Reviewer 4 requests that the language of the manuscript has to be checked one more time since there are some misuses of the English words. The manuscript has been re-read and checked by the native English-speaking person. As such, the word “overlook” has been changed to “overview” (Page 2), word “signalization” to “signaling” (Page 2 and 6), word “actors” to “players” (Page 2), etc.
- The Reviewer 4 suggests an alternative title to our manuscript. In fact, the suggested by the Reviewer 4 title doesn’t contain any notion nor link between Ca2+-signaling and therapeutic targeting in PDAC, while in our manuscript the whole chapter 4 is dedicated to the role of Ca2+-signaling in PDAC treatment. This chapter represents one third of the manuscript and considers in detail different compounds and treatment options in PDAC and their cross-talk/targeting of Ca2+-signaling. That is why to our opinion “…and its potential targeting…” should be represented in the title as it is in the current form.
- The Reviewer 4 suggests the deletion of the introductory paragraph on the ion channels. Actually, according to our experience, not all if only few readers who are familiar with the ion channels notions. Since one of the “Cancers” journal policies is that the manuscript has to be dedicated to the broad readership, we have adhered to this policy and introduced them. Moreover, all three other Reviewers have pushed us to provide detailed introductions to the Ca2+-signaling, Ca2+-transporters, etc., we believe that a small introduction to ion channels could be a useful feature of our manuscript.
- The Reviewer 4 suggests the deletion of Table 1 since it is not necessary because is well explained in the manuscript. However, this Table is a succinct representation of all calcium transporters (and their isoforms) reported so far in PDAC, their expression, localization, and the potential role. If you see the corresponding text not all of these features are indicated, listed, and described. Therefore, our Table 1 completes and summarizes the data of the chapter 2. It’s like any Figure, which seems to be redundant to the text but actually is complementary and crucial for the visual representation of the data. We wish to leave it since it represents clear benefits for the readership, rather than any drawback.
- The Reviewer 4 indicates some plagiarism suspicion. We would like to assure the Reviewer 4 that the entire manuscript has been checked on the plagiarism using an on-line available software (e.g.: https://www.duplichecker.com/) before the initial submission. We believe that nowadays all the manuscripts are plagiarism-checked before any reviewing step further. For the paragraph in question, we have checked it again and the software gives us 0% plagiarism, 100% unique. Nevertheless, this sentence has been changed as: “By reversing the PDAC cells metabolism while depriving cells of glucose and supplementing with α-ketoisocaproate or galactose (substrates that promote mitochondrial metabolism), the dependence of high glycolytic PDAC cells and in particular the glycolytic dependence of the PMCA, has been revealed. This feature could be exploited therapeutically to treat PDAC”. (Page 4).
- The Reviewer 4 suggests the authors include the full form of a molecule along with the abbreviation when mentioned for the first time. That is actually what we did from the very beginning (NCX was abbreviated on the page 4, and MCU it was right in the Table 1 above). Nevertheless, we have checked for the abbreviations again to ensure that everything is clear for the readers.
- The Reviewer 4 does not see any difference between Figures 1 and 2, and believes that Figure 1 is redundant. Actually, while having a closer look one can see that Figure 1 as its title suggests: “Calcium transporters in Pancreatic ductal adenocarcinoma (PDAC)” represents only the schematic representation of all kind of calcium transporters published so far in PDAC as well as their subcellular localization. This Figure 1 intends to help the reader to localize the transporters in the subcellular context. The Figure 2, as its title suggests: “Calcium signaling is involved in the cell proliferation and apoptosis in PDAC”, is meant to present mostly Ca2+-signaling pathways and only Ca2+-transporters involved in proliferation and apoptosis resistance, rather than all the Ca2+-transporters reported so far. As such, these Figures cannot be merged to avoid a mess and to worsen readers’ comprehension.
- Finally the last comment of the Reviewer 4 concerns the review structure and completeness to meet the beforehand assigned task (before writing this review, the abstract and the table of contents were sent to the Editors of “Cancers” for the approval). We do agree that as such, we have a full almost systemic review accompanied with one Table and 4 Figures which can be hardly considered as a concise review. Despite the number of data on the Ca2+-transporters and Ca2+-signaling pathways is still very limited, it was of our primary goal to gather all these data and to present them in the comprehensive reader-friendly form emphasizing that Ca2+-transporters are prospective therapeutic targets.
Moreover, the Reviewer 4 claims “the article only suggested what pumps or channels could play a potential role in the development of PDAC and urged the community to investigate them”. We apologize but there is no one Ca2+-transporter mentioned in our review which was not reported in the PDAC (please see Table 1).
Further, the Reviewer 4 implies that we do not show the role of these same Ca2+-transporters in other type of cancers (being rather critical to the broadness of our review at the same time). In fact, one can found 5 comparisons with the breast cancer, 4 comparisons with the glioblastoma, 3 with the prostate cancer, 1 ovarian, etc…
In conclusion, we are grateful to the Reviewer 4 as well as to all other 3 Reviewers to help us to improve our manuscript which we strongly believe will be a timely and solid (even if broad) contribution to the field. Thank you!
Round 2
Reviewer 4 Report
We would like to thank the authors for their point-by point defense against our critics. We have come to agree to most of the responses. Having said that, we still think the length of the article could have been shortened. But that we would leave to the editors.
Last but no least, we would still suggest the author to provoke new ideas/suggestions to the scientists who in future, would work on calcium signaling and it's targets in the field of PDAC.